# DNA Methylation Profiles of *GAD1* in Human Cerebral Organoids of Autism Indicate Disrupted Epigenetic Regulation during Early Development

**DOI:** 10.3390/ijms23169188

**Published:** 2022-08-16

**Authors:** Georgina Pearson, Chenchen Song, Sonja Hohmann, Tatyana Prokhorova, Tanja Maria Sheldrick-Michel, Thomas Knöpfel

**Affiliations:** 1Laboratory for Neuronal Circuit Dynamics, Imperial College London, London W12 0NN, UK; 2Department of Psychiatry, Department of Clinical Research, University of Southern Denmark, 5230 Odense, Denmark

**Keywords:** DNA methylation, epigenetics, autism, organoids, GABA, *GAD1*, CTCF

## Abstract

DNA methylation profiling has become a promising approach towards identifying biomarkers of neuropsychiatric disorders including autism spectrum disorder (ASD). Epigenetic markers capture genetic risk factors and diverse exogenous and endogenous factors, including environmental risk factors and complex disease pathologies. We analysed the differential methylation profile of a regulatory region of the *GAD1* gene using cerebral organoids generated from induced pluripotent stem cells (iPSCs) from adults with a diagnosis of ASD and from age- and gender-matched healthy individuals. Both groups showed high levels of methylation across the majority of CpG sites within the profiled *GAD1* region of interest. The ASD group exhibited a higher number of unique DNA methylation patterns compared to controls and an increased CpG-wise variance. We detected six differentially methylated CpG sites in ASD, three of which reside within a methylation-dependent transcription factor binding site. In ASD, *GAD1* is subject to differential methylation patterns that may not only influence its expression, but may also indicate variable epigenetic regulation among cells.

## 1. Introduction

The methylation of DNA at distinct nucleotides, most commonly cytosines followed by a guanine residue (CpG site), is an essential epigenetic modification that provides a reversible mechanism for regulating gene expression without altering the DNA sequence [1]. During embryonic development, DNA methylation plays an essential role in regulating gene expression and in the differentiation of cells towards particular lineages [2]. Methylation signatures can be influenced by various environmental factors, including maternal stress, drugs, and diet; hence, they reflect an integration of genetic and environmental influences [3,4]. As the methylation status of a particular gene can be sustained throughout an organism’s lifespan, it may represent a molecular-level memory of developmental processes, which include any environmental impacts. Consequently, DNA methylation is an important molecular control mechanism through which brain functions are established and maintained.

Autism spectrum disorder (ASD) is a pervasive neurodevelopmental disorder characterised by challenges in social interaction and communication, as well as repetitive or restrictive behaviours or interests [5]. Various risk factors, both genetic and environmental, have been identified as possible contributors to the development of ASD [6,7,8]. Abnormalities in the epigenome and its regulatory components have been associated with ASD, in some cases playing a key role in the progression of the disorder [9], and genome-wide analyses have revealed individual CpG sites subject to differential DNA methylation between patients and controls [10,11,12]. Such changes have also been identified in the placental tissue [10], suggesting an epigenetic influence during the embryonic neurodevelopmental process that may serve as a molecular control mechanism with functional relevance in circuit neurophysiology.

The balance between excitation (E) and inhibition (I) is a key principle for neuronal network organisation and information processing [13,14,15]. Consistent with this notion, excitation–inhibition imbalances are considered a pathophysiological mechanism in many brain disorders including ASD [16,17,18,19]. As the primary modulator of inhibitory neurotransmission, γ-aminobutyric acid (GABA) plays a key role in the maintenance of an appropriate E/I balance within the brain. A key gene in the regulation of GABA levels is *GAD1*, which encodes a glutamic acid decarboxylase enzyme isoform that catalyses GABA synthesis and is an excellent biomarker in the assessment of E/I balance at the molecular level.

It is clear that there is evidence for epigenetic dysregulation being implicated in the imbalance of E/I in ASD [20,21], and that this may be closely related to the severity of key symptoms. What remains a challenge, however, is integrating the findings from the brain-tissue-specific post-mortem analyses of patients with evidence from rodent models of ASD, indicating that *GAD1* DNA methylation patterns are disturbed during development [22,23]. To investigate how the *GAD1* methylation pattern during embryonic development may differ in ASD, we generated cerebral organoids from iPSCs derived from individuals with and without ASD, and performed targeted DNA methylation profiling of a known regulatory target region of the *GAD1* gene [24].

## 2. Results

### 2.1. GAD1 DNA Methylation Is More Diverse in Cerebral Organoids from ASD Subjects

We profiled the methylation status of a regulatory region of the *GAD1* gene approximately 3400 bp downstream of the transcription start site (TSS) (Figure 1) by bisulfite sequencing using DNA extracted from whole cerebral organoids produced from iPSCs derived from either control or ASD subjects. We chose this CpG island region proximal to the *GAD1* promoter, as it was shown to have a functional relevance to GABA expression and includes a CpG-containing CTCF binding site that may contribute to its regulation [24]. The cerebral organoids used in this study were derived from 5 control and 4 ASD subjects. The cell lines used to grow these organoids have been characterised in detail in a recent paper, and representative images of these organoids are presented in Figure 1 by Ilieva et al. [25].

We observed distinct methylation profiles within the control samples, where the majority (94.3%) of the randomly picked clones had high methylation levels throughout the amplicon. Clones with <50% methylation across the amplicon were only detected in one out of the five control organoids (Figure 2A). In the ASD dataset, although the vast majority (90.3%) of clones also exhibited high levels of methylation, three of the four ASD organoids contained patterns with <50% methylation across the amplicon (Figure 2B). In the following, we first limit our analysis to the dominant >50% methylation groups.

Strikingly, the diversity in DNA methylation profiles across the amplicon was increased in the ASD group. To compare this methylation pattern diversity, we analysed the occurrence of distinct patterns in each of the dominant >50% methylation populations. While the control group exhibited only five distinct patterns (2 distinct patterns in organoids from control subjects #1–#4), sixteen patterns were observed from the ASD group (Figure 3). In line with this, patterns 1 and 2 alone accounted for 96% of control-derived clones but only 73% of ASD-derived clones. The most common pattern seen in both groups, pattern 1, was that of near complete methylation, with only CpG sites 18 and 28 being unmethylated (69.7% of control-derived clones; 46.4% of ASD-derived clones). Pattern 2 consisted of the methylation at all 39 CpG sites and occurred in approximately 25% of clones in each group. The remaining patterns occurred with minimal frequency, often appearing in only one clone. One signature, pattern 5, was unique to organoid #5 of the control group, whereas patterns 6–17 were all seen in the ASD subject population only. Note that patterns 7 and 8 are almost identical to pattern 1, in both cases with one of the CpG sites presenting a C-to-A mutation.

For the much smaller group of rarer clones with <50% methylation, sample numbers are too small for a detailed analysis as above. However, the percentage of CpG sites within the amplicon that are methylated is lower in the control group as compared to the ASD group (3.85 ± 2.22 % versus 14.53 ± 4.06 %, mean ± SEM, N = 4 and 6 clones respectively, *p* = 0.052, one-sided Wilcoxon rank sum test).

### 2.2. Differential Methylation of Specific CpG Sites

To further compare the diversity of DNA methylation signatures, we analysed the methylation variability of each CpG site (for clones with >50% methylation across the amplicon). We detected a total of six CpG sites that are significantly differentially methylated in ASD relative to the control group (Figure 4A; CpG site 23, *p* = 0.002; site 28, *p* = 0.017; site 29, *p* < 0.001; site 30, *p* = 0.015; site 34, *p* = 0.028; site 35, *p* = 0.028; site 39, *p* = 0.028; Wilcoxon rank sum; N = 56 clones from ASD group, 66 clones from control group), and an increased overall CpG-wise variability in ASD (Figure 4B; *p* < 0.001 Wilcoxon rank sum). Five out of the six differentially methylated CpG sites are hypomethylated in ASD. Notably, three of these differentially methylated CpG sites lie within a CTCF binding site and the remaining sites are in close proximity (within 100 base-pairs up- and downstream), suggesting that this regulatory region is prone to differential methylation in ASD.

## 3. Discussion

GABAergic signalling is crucial for the excitation–inhibition balance in neural circuits, which is often disrupted in neuropsychiatric disorders such as ASD [32,33]. While genome-wide analyses have successfully identified a number of genes subject to differential methylation between adult controls and patients, and have even indicated disruption amongst complex networks of associated genes [34,35], whether epigenetic signatures are different in the neurodevelopment of normal and ASD brains has long been postulated but remains unknown. To address this, here we focused specifically on a functionally relevant regulatory region of *GAD1* and observed distinct differences in the DNA methylation profiles of cerebral organoids derived from ASD and control subjects. Our results identified specific CpG sites that were subject to hypo- and hypermethylation in ASD in regions that may be crucial to the regulation of gene expression. Furthermore, by analysing the frequency with which the methylation patterns occur, we identified a loss of regularity in the methylation profile of *GAD1* in organoids derived from ASD patients.

The advancement of organoid technologies has enabled significant progress in the understanding of neurodevelopmental processes and disorders. Already cerebral organoids have been used in drug screening and to validate suspected ASD risk factor genes [36,37,38,39]. We used human cerebral organoids (d39) that were developmentally equivalent to brains at 11–12 weeks post-conception [40]. Similarities in the morphology, physiology, and transcriptome suggest that the in vitro model recapitulates the developmental process to a high degree, particularly at this time point [41,42,43]. Further, the epigenome of cerebral organoids closely resembles that of the human developing brain, and *GAD1* itself shows little to no differential methylation between the two [44]. Therefore, we are confident that the methylation signatures presented in this study are representative of those seen during embryonic development.

Methylation signatures vary not only between individuals, but also in a tissue- and cell-type dependent manner [45]. We, therefore, subcloned our target amplicon in order to identify *GAD1* DNA methylation patterns with single-cell resolution, thereby exposing the uniformity of each group. Among those that exhibited >50% methylation, 11 signatures were exclusive to single clones within the ASD group, demonstrating that a high proportion of cells possessed unique methylation patterns. In contrast, in the control group only one clone exhibited a unique methylation pattern. The high level of variability within the ASD group and the increased CpG-wise variance suggests disruption within the components that regulate DNA methylation during development. In adults diagnosed with ASD, the mRNA expression of key components of the epigenetic machinery, namely ten-even translocase (TET) enzymes, is significantly increased in the brain [21]. Moreover, DNA methyltransferase 1 (DNMT1) binds with increased frequency to *GAD1* in a region upstream of our target region [21]. DNMT is responsible for maintaining DNA methylation through cellular division, and the TET family of enzymes mediate active demethylation [46,47]. Therefore, we assume that disruptions in both epigenetic components may have early developmental origins, thereby producing the increased diversity of methylation patterns seen in the ASD group.

Within our region of interest resides a binding motif for CTCF, a transcriptional factor protein able to regulate transcription through its interactions with enhancers and promoters. The binding sequence present in our region of interest (GTTGC**CGCGCG**GGGG**CG**CTTT) contains four CpG sites and is known to be sensitive to DNA methylation [24]. When methylated, CTCF is unable to bind to the motif, resulting in increased *GAD1* transcription and mRNA expression [24]. Of the four CpG sites within this sequence, three were subject to significantly differential methylation in the ASD group. Despite being consecutive CpG sites in the sequence, site 28 was hypermethylated while sites 29 and 30 were hypomethylated. This was curious given that neighbouring CpG sites often share the same methylation status [48]. The differential methylation of these three sites indicates an altered CTCF binding probability in organoids derived from ASD subjects, although the functional significance of the methylation status of each individual CpG site on the sensitivity of the CTCF binding is unknown. In line with the methylation dependency of the CTCF binding, the overall trend towards decreased methylation within the binding sequence would suggest that *GAD1* transcription is decreased in ASD organoids. This would, therefore, corroborate the decreased expression of GAD1 and GAD-67 that is seen in ASD patients [21,49,50].

DNA methylation can vary in a cell-type-specific manner. In the brain, the methylation signatures of a particular gene can differ between neuronal subpopulations, such as between excitatory and inhibitory neurons [51,52]. The ASD patient-derived cerebral organoids also indicate cell-type-specific disturbances between these two neuronal populations, indicating a relevance in ASD progression [37,53]. In light of its role in GABAergic interneurons, it would be reasonable to assume that *GAD1* may display distinct methylation patterns specific to this cell type. Therefore, future endeavors should incorporate cell-type-specific methylation profiling to provide further insight into the functional influence of DNA methylation and targeted epigenetic dysregulation during neurodevelopment in ASD.

In recent years, the development of organoid cultures has provided an unparalleled opportunity to vastly improve the translatability and relevance of neuroscience studies. Derived from human stem cells and iPSCs, 3D organoids circumvent many of the issues associated with animal models or traditional clinical methods [54]. While the potential of human-derived organoids is remarkable, the validity of the epigenetic signatures exhibited must be scrutinised. Before differentiation, mouse fibroblast iPSCs can be distinguished from embryonic stem cells by analysing differential methylation patterns, suggesting a maintained methylation memory of the tissue of origin [55]. Although this appears to be relevant to genes associated with gene expression and pluripotency, this does highlight a need for caution when translating the findings from cerebral organoids into conclusions about the human brain.

A large body of evidence points to the disruption of early embryonic development as a crucial factor for the manifestation of ASD, meaning identifying early deviations in neural development is key to understanding ASD. Our study contributes to a plethora of evidence that implicates *GAD1* in autism, but adds unique knowledge on the role of *GAD1* at the embryonic development stage of the condition and indicates that the gene is subject to altered epigenetic regulation in the early stages of neurodevelopment. The identification of six potential GABAergic biomarkers, three of which lie within the functionally relevant CTCF binding sequence, may provide a link between epigenetic disruption and the altered balance of excitation and inhibition in autism.

## 4. Materials and Methods

### 4.1. iPSCs and Generation of Cerebral Organoids

Reports on the derivation of iPSCs from recruited subjects and the generation of cerebral organoids were published previously [25,28,29,30,31]. In summary, four individuals with idiopathic autism spectrum disorder (ICD 10: F84.1–84.8, male; mean age of 21 years) and five age- and gender-matched healthy controls were included in the study. Patients diagnosed with ASD were randomly recruited from the Fyn Autism Cohort described in [56]. They all fulfilled the diagnostic criteria for developmental disorder according to ICD 10 [57], which translates to autism spectrum disorders in the DSM V [5]. The age-matched controls were selected after an advertisement on a local blackboard and social media.

Fibroblasts derived from skin punch biopsies taken from ASD subjects and age-matched controls were reprogrammed into iPSCs with episomal plasmid vectors containing Oct3/4, Sox2, Klf4, LIN28, and L-Myc (pCXLE-hUL-L-MYC, LIN28, Addgene#27080, pCXLE-hSK-SOX2, KLF4, Addgene #27078, and pCXLEhOCT4-shp53-F-OCT4, shP53, Addgene #27077) according to the protocol previously described by Ilieva et al. and Kamand et al. [25,28,29,30,31], and the reprogramming efficiency and quality of the derived iPSCs were evaluated using standardization criteria [28,29,30,31].

The cerebral organoids were generated according to the protocol described by Lancaster and Knoblich [58] and modified by Ilieva et al. [25], with a Stemdiff™ Cerebral organoid kit (Stemcell Technologies, #08570). The cerebral organoid formation was initiated through an embryoid body (EB) formation. The iPSCs were collected and seeded at a density of 9000 cells/well in a 96-well round-bottom ultra-low attachment plate (Corning) in EB formation medium supplemented with ROCK inhibitor. On days 5–7, the neuroectoderm formation was induced by moving the EB into induction media in ultra-low attachment 24-well plates. On day 7, the EB was embedded in Matrigel (Corning) droplets and cultivated in an expansion medium in a six-well ultra-low adherence plate. At day 10, the embedded organoids developed expanded neuroepithelia, as evidenced by budding of the surface. In this stage, they were moved to a maturation medium and cultivated on an orbital shaker. The genomic DNA was extracted using an AllPrep Kit (Qiagen) from one organoid per subject at day 39 of development.

### 4.2. Bisulfite Conversion and PCR Amplification

The genomic DNA was bisulfite converted using EpiTect Bisulfite Kit (Qiagen), then amplified using HotStar Taq Plus Master Mix (Qiagen). The target region and primer sequences are shown in Figure 1. The total length of the profiled *GAD1* amplicon was 560 bp. The following PCR conditions were used: 95 °C for 5 min; 40 cycles of 94 °C for 1 min, 52 °C for 1 min, 72 °C for 1 min; 72 °C for 30 min.

### 4.3. TOPO Cloning

PCR products were subcloned using a TOPO™ TA Cloning™ Kit with One Shot™ TOP10 chemically competent E. coli cells (Invitrogen). Up to 20 colonies per organoid were picked after blue-white selection. The DNA was purified using QIAprep Spin Miniprep Kit (Qiagen) and used for the Sanger sequencing (Genewiz UK).

### 4.4. DNA Methylation Analysis

The trace files were analysed using the QUMA online tool, available at http://quma.cdb.riken.jp/ (accessed 30 April 2022) [59]. The lollipop plot figures were produced using the Methylation Plotter web tool, available at http://gattaca.imppc.org/methylation_plotter/ (accessed 30 April 2022) [60]. The differential methylation at each CpG site was analysed in MATLAB and plotted as the (average percentage methylation in ASD group) minus (average percentage methylation in control group) for each CpG site (as such, positive values indicate hypermethylation in the ASD group relative to the control).

### 4.5. Statistical Analysis

The binary clone sequence data from different organoids of the same condition were pooled for the statistical analysis. The statistical analysis was performed using MATLAB with Statistics and Machine Learning Toolbox. For the CpG-wise differential methylation, Wilcoxon signed rank test was performed for each CpG site between the control vs. ASD conditions using the MATLAB command *signrank*. For the statistical test on the amplicon-wide methylation differences (for <50% methylated clones), we calculated the percentage of methylated CpG sites across the amplicon and performed a one-sided Wilcoxon rank sum test between the control and ASD conditions using the MATLAB command *ranksum*.

## Figures and Tables

**Figure 1 ijms-23-09188-f001:**
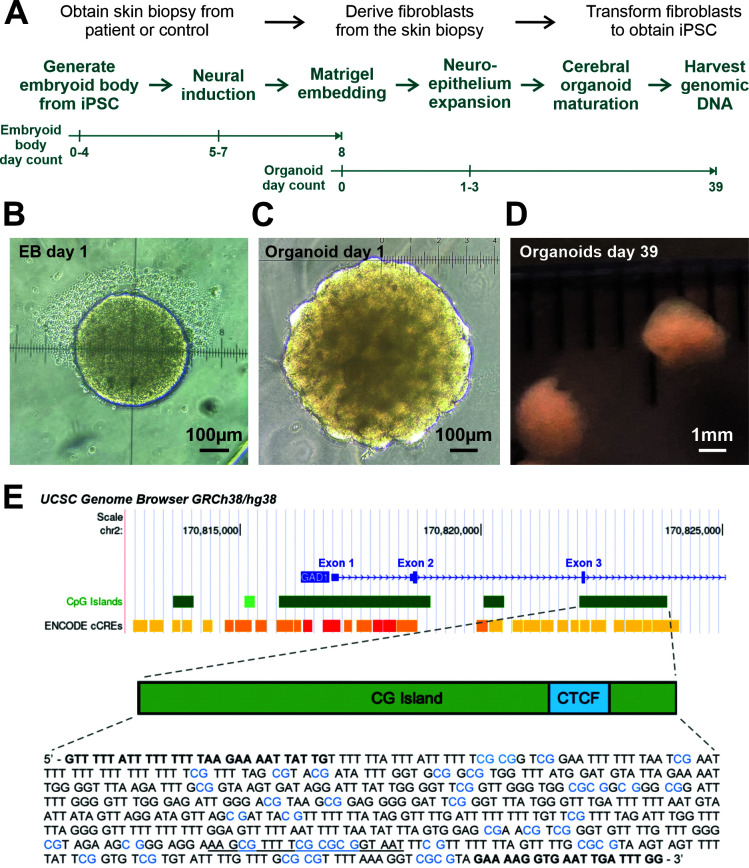
Organoid production stages and human *GAD1* gene target region. (**A**) The production of human-derived organoids follows two main stages. Upper: Using subject skin biopsies to generate human inducible pluripotent stem cells. Lower: Using the inducible pluripotent stem cells to produce the embryoid body, which will then develop into brain organoids. (**B**) Example image of an embryoid body at embryoid body (EB) day 1. (**C**) Example image of a brain organoid at organoid day 1. (**D**) Example image of brain organoids at organoid day 39, when the genomic DNA is harvested for *GAD1* methylation profiling. (**E**) The human *GAD1* target region for the methylation analysis mapped onto the UCSC genome browser (assembly GRCh38/hg38, https://genome.ucsc.edu/ (accessed on 30 April 2022) [26,27]). We profiled a CTCF binding-site-containing CpG island that has been shown to correlate with *GAD1* gene transcription in human cells [24]. Sequence shown is specific to bisulfite-converted DNA, with the 39 sites of potential CpG methylation highlighted in blue. Underlined region denotes a CTCF binding sequence. Primer annealing regions used to amplify the region of interest are indicated in bold. Additional images and characterisation of subject-derived organoids can be found in [28,29,30,31].

**Figure 2 ijms-23-09188-f002:**
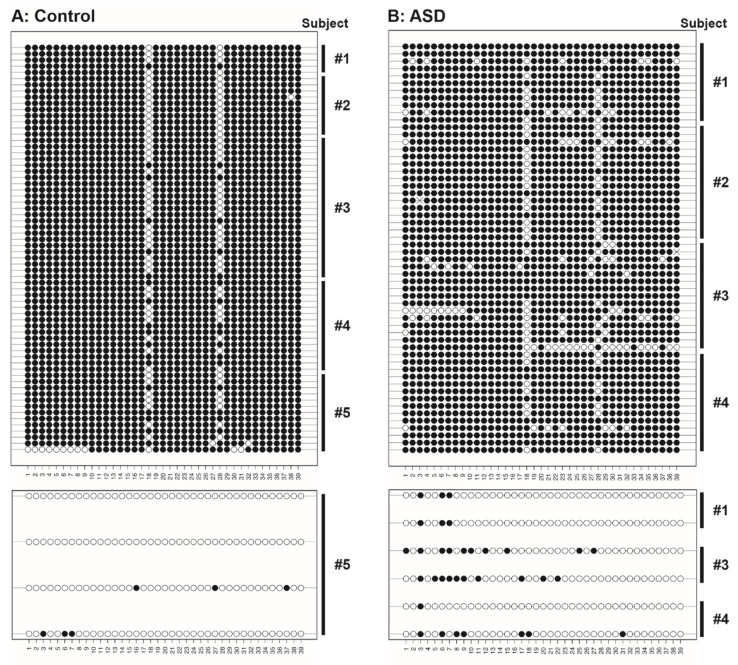
Lollipop plot of methylation patterns of individual CpG sites across the *GAD1* target region of interest in cerebral organoids derived from (**A**) control (N = 70 clones from 5 subjects) and (**B**) ASD subjects (N = 62 clones from 4 subjects). Rows represent the *GAD1* amplicons from randomly picked individual clones, symbols represent individual CpG sites within the amplicon. Filled dots, methylated cytosines; empty dots, unmethylated cytosines; ×, cytosine-to-adenine mutation. Clones with <50% methylation are shown separately on the lower panels.

**Figure 3 ijms-23-09188-f003:**
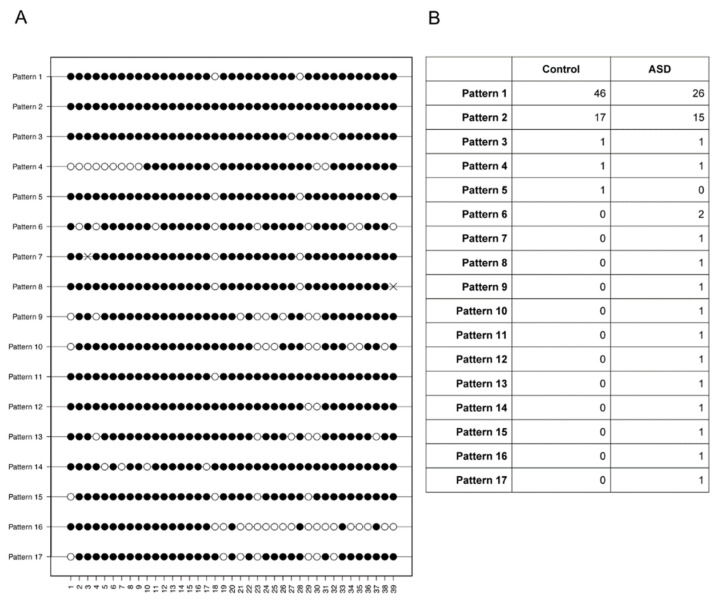
DNA methylation pattern diversity in the *GAD1* region of interest in cerebral organoids. (**A**) Distinct patterns of DNA methylation signatures in the two >50% methylation groups. Filled dots, methylated cytosines; empty dots, unmethylated cytosines; ×, cytosine-to-adenine mutation. (**B**) The number of clones within the ASD and control groups displaying each DNA methylation pattern.

**Figure 4 ijms-23-09188-f004:**
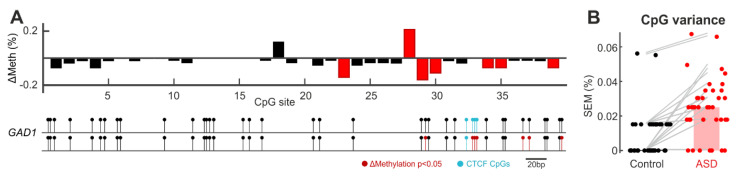
(**A**) Upper: CpG-wise differential methylation across the profiled *GAD1* region of interest. Differential methylation plotted as (average percentage methylation in ASD group) minus (average percentage methylation in control group) for each CpG site, such that positive values reflect hypermethylation in ASD relative to controls. N = 56 clones from ASD group; 66 clones from control group. Red bars: *p* < 0.05 Wilcoxon signed rank. Lower: CpG sites in the profiled *GAD1* amplicon mapped onto the human genome. (**B**) Increased CpG methylation variance in ASD. Bar height: Median variance (SEM) at each CpG site (N = 39 CpG sites, *p* < 0.05 Wilcoxon rank sum. Note: Median variance for control group is 0). Grey: Corresponding variance between control and ASD at each CpG site.

## Data Availability

All data used are shown in the manuscript. The tabulated format is available from the corresponding author upon request.

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
