# Peer review of "DNA Methylation Profiles of *GAD1* in Human Cerebral Organoids of Autism Indicate Disrupted Epigenetic Regulation during Early Development"

_ijms, 2022, doi:10.3390/ijms23169188_

Round 1
Reviewer 1 Report
Pearson et al reported DNA methylation profiles of GAD1 in human cerebral organoids of autism and found disrupted epigenetic regulation during early development. The authors analysed the differential methylation profile of a regulatory region of the GAD1 gene using cerebral organoids. The authors found high levels of methylation across the majority of CpG sites within the GAD1 region in both groups. Moreover, ASD group exhibited a higher number of unique DNA methylation patterns compared to controls. This work is interesting. However, there are some concerns that the authors should address before publication.
Comments:
1. In Figure 1, the authors should make a schematic illustration of the GAD1 gene containing the promoter and exon1 locations.
2. The authors should label the location of the detection region of GAD1 by DNA methylation analysis.
3. The description of statistical analysis should be in more detail, including all statistical methods used in this study.
4. The ASD organoids are very good models for studying the molecular mechanism of ASD, thus the authors should display representative images of organoids used in this study, and also show the diagram of the process of establishing ASD organoids, which can benefit the readers.
5. Although the authors analyzed DNA methylation pattern diversity in the GAD1 region, the authors did not provide evidence of which pattern is more related to GAD1 expression at transcriptional level. The authors should identify the core methylation regulatory pattern in the GAD1 promoter region.
6. The authors should examine GAD1 expression at transcriptional level in ASD organoids and controls.
Reviewer 2 Report
The work from Pearson et al. describes the analysis of the methylation pattern in cerebral organoids derived from control and ASD patient’s iPSC. Because of the importance of the GAD1 gene in excitation/inhibition balance in neural network and as key regulator of GABA synthesis the Authors investigated its methylation profile. They found that CpG sites were both subjected to hypo- and hyper-methylation. Among the ASD group exhibiting >50% methylation there was a high level of variability and different pattern compared to controls. There is still little known about the role of epigenetics during development in ASD and this work provides useful information. There are however several points to be addressed.
Specific points
Results
The authors analyzed the methylation status of cortical organoids at 39 days of culture. Representative images of organoids used for the analysis should be provided. Are there differences in development of organoids from control and ASD patients? It would be useful to show immunofluorescence/wb for differentiation or stemness markers to see whether methylation changes are associated with developmental defects.
The work would be really improved if the authors could show the level of expression of GAD1 gene to correlate it with the methylation disturbances and whether they could show the interaction between the regulatory region of GAD1 and the CTCF transcription factor.
Discussion
There are several papers describing the identification of methylation markers potentially useful for ASD diagnosis. These references should be added and discussed to the discussion section (e.g DOI: 10.1007/s11011-021-00805-5; DOI: 10.3390/ijms21186877)
Information regarding GAD1 expression in ASD patients (that was reported as decreased) should be commented in discussion.
Previous works on ASD organoids also should be cited and commented in the discussion (e.g doi: 10.1038/s41467-021-24358-4.)
Round 2
Reviewer 1 Report
The authors have addressed my concerns, and I recommend publication in the journal.
Reviewer 2 Report
The Authors have addressed all the concerns and the paper is acceptable for publication